# Bochum Assessment of Avoidance-based Emotion Regulation for Children (BAER-C): Development and evaluation of a new instrument measuring anticipatory avoidance-based emotion regulation in anxiety eliciting situations

Michael W. Lippert[1]*, Katharina Sommer[1], Tabea Flasinski[1], Jan Schomberg[1], Verena Pflug[1], Hanna Christiansen[2], Tina In-Albon[3], Susanne Knappe[4], Marcel Romanos[5], Brunna Tuschen-Caffier[6], Silvia Schneider[1]

1 Mental Health Research and Treatment Centre, Ruhr University Bochum, Bochum, Germany, 2 Clinical Child and Adolescent Psychology, Phillips-University Marburg, Marburg, Germany, 3 Clinical Child and Adolescent Psychology and Psychotherapy, University of Koblenz-Landau, Landau, Germany, 4 Institute of Clinical Psychology and Psychotherapy, Technische Universität Dresden, Dresden, Germany, 5 Department of Child and Adolescent Psychiatry, Psychosomatics and Psychotherapy, University Hospital Würzburg, Würzburg, Germany, 6 Clinical Psychology and Psychotherapy, University Freiburg, Freiburg, Germany

* michael.lippert@rub.de

## Abstract

Avoidance-based emotion regulation plays a central role in the development and maintenance of anxiety disorders across the life span. However, measures for children that account for different avoidance strategies, are scarce. Derived from Gross' Process Model of Emotion Regulation, the Bochum Assessment of Avoidance-based Emotion Regulation for Children (BAER-C) was developed to assess avoidance strategies (cognitive avoidance, behavioural avoidance, verbal reassurance, and social reassurance) and reappraisal in anticipatory anxious situations. In the present study, the BAER-C was administered to 129 school children aged 8 to 14 and 199 children with anxiety disorders aged 8 to 16 and their parents, along with established measures on anxiety, psychopathology, and emotion regulation. Factor structure, internal consistency, convergent, divergent and construct validity were analysed. Results of the anxious sample showed a satisfactory internal consistency (McDonald's $\omega$ = .94) for all scales as well as positive correlations with anxiety symptoms (all $rs$ > .17, all $ps$ < .05). Factor analysis supported a five-factor model. This model was confirmed in the student sample. Children with an anxiety disorder scored higher on behavioural avoidance, verbal reassurance, and social reassurance than school children ($F$ (5,304) = 12.63, $p$ = .003, $\eta_p^2$ = .17). Results for construct validity were ambiguous. Our analyses suggest that the BAER-C is a promising theory-based new instrument to reliably assess different avoidance strategies in children. More research is needed to further analyse construct validity with other emotion regulation questionnaires.

**Data Availability Statement:** Data cannot be shared publicly because they contain highly confidential health data, which might allow identification of individuals in a vulnerable patient group (children). Data can be requested by contacting Dr. Xiao Chi Zhang (Xiaochi. Zhang@ruhr-uni-bochum.de) and the Protect-AD P2 consortium.

**Funding:** This study was supported by the research consortium on child anxiety disorders, PROTECT-AD, P2, awarded to SS funded by the German Federal Ministry of Education and Research https://www.bmbf.de (FKZ 01EE1402C). The funders had no role in study design, data collection and analysis, decision to publish, or preparation of the manuscript.

**Competing interests:** The authors have declared that no competing interests exist.

# 1 Introduction

Anxiety disorders (ADs) show period prevalence rates of 6% to 20% and are thus highly prevalent amongst children and adolescents [1–3]. Childhood anxiety disorders (CADs) are categorized by an excessive pattern of fear or anxiety towards a certain situation or object [4]. They cause substantial impairment for the patient and their family and are a risk factor for healthy development [5,6]. In addition, CADs pave the way for the development of other disorders (anxiety, depression, substance abuse etc.) in adolescence and adulthood [7,8]. Cognitive behavioural therapy (CBT) has proven to be a reliable and effective treatment for CADs [9–11] and has positive effects on general functioning and depressive symptoms [11]. Even so, 20% to 40% of patients do not respond to standard CBT [12].

One reason for the high number of non-responders could be the sparse theoretical foundation and the lack of basic research in the field of CADs. It is vital to identify and understand the key components and mechanisms involved in the development and maintenance of CADs in order to improve children's anxiety treatment. One hot candidate, also described as a transdiagnostic factor in the etiology across anxiety and other mental disorders, is emotion regulation [13,14]. Dysfunctional emotion regulation could be partially responsible for the elicitation as well as the maintenance of a broad range of mental disorders, including CADs [15]. Emotion regulation (ER) is defined as the modification of intrinsic or extrinsic processes, which leads to emotions or emotional manifestation in behaviour [16]. These processes are specified in numerous ER strategies [e.g., suppression, reapprasial; 17]. Gross [18] describes ER as a process in which the modification of an emotion can occur at different stages: first in the selection or modification of the situation, followed by the deployment of attention and cognitive mechanisms of appraisal, and finally in the modulation of the emotional response. Gross [18] proposes that ER strategies can be sorted into five categories aligned with the stages in the ER process: situation selection, situation modification, attentional deployment, cognitive change, and response modulation.

One ER strategy, namely avoidance, has been identified to play a role in the mechanism of the most prominent models of CADs. An early definition of avoidance describes avoidance as a process, in which the organism learns to do "whatever type of adjustment to eliminate [fear inducing] stimuli or attendant fear" [19]. Modern definitions expand from fear to more general unpleasant or painful situations that a person has learned to anticipate and therefore to avoid [20].

In the short run, avoidance is highly effective in reducing negative emotions (e.g., anxiety), however, it prevents the individual from making corrective learning experiences that could reduce anxiety in the long run [21,22]. Avoidance can manifest itself on a behavioural and cognitive level in different types of strategies [23]: 1) Behavioural avoidance (e.g., changing the side of the street if a dog approaches) is a classical behavioural strategy observed in CADs. However, the research focus here is on specific phobia [23]. 2.) Safety behaviour (e.g., wearing wide clothes, trying not to be the centre of attention or seeking parents' assurance) is a behavioural strategy commonly used by children and adolescents with social anxiety disorder (SAD), generalized anxiety disorder (GAD), and separation anxiety disorder (SepAD) [24]. 3.) Thought suppression (e.g., trying to ignore anxiety invoking thoughts) is a cognitive-focused avoidance strategy commonly used by children with SAD or GAD [25,26].

Research to assess avoidance as an ER strategy is rather scarce in spite of its prominent role in the mechanisms of anxiety disorders. Avoidance can be assessed either with behavioural focused approaches or with psychometric instruments. Behavioural tests with either observational measures, like the Behavioural Approach Task [BAT, e.g., 27,28], eye-tracking measurement [29] or implicit measures, like the Approach Avoidance Task [AAT; 30], use the actual

presence of the fearful stimulus or pictures of it. These tasks require additional technological equipment and are rather costly in their implementation [31]. As an alternative, psychometric instruments can be easily and efficiently used in routine treatment. Most of the existing anxiety questionnaires or diagnostic interviews contain avoidance related items [e.g., SAAI-C/P, 32]. However, these items are rather unspecific and often focus only on the quantity of avoidance behaviour in disorder-specific situations [33,34]. Furthermore, ER questionnaires usually do not measure avoidance as a specific strategy.

One exception is the Child Avoidance Measure-Self Report and Child Avoidance Measure-Parent Report [CAMS and CAMP, 31] that measures behavioural avoidance in CAD. However, CAMS/CAMP sets focus on behavioural avoidance and does not include other avoidance strategies. The questionnaires show good psychometric properties as well as high correlations with anxiety symptoms. The German language ER questionnaire, the FEEL-KJ ["Fragebogen zur Erhebung der Emotionsregulation bei Kindern und Jugendlichen", 35], measures a total of 15 different ER strategies. Each ER strategy is measured with two items for the three emotions anxiety, sadness, and anger. Some of the strategies can be sorted into a two-factor model of adaptive and maladaptive strategies. The questionnaire follows a bottom-up approach with no theoretical model as a foundation. Although the questionnaire covers a broad range of 15 different ER strategies, it does not assess avoidance. When Braet et al. [36] used the FEEL-KJ to connect different ER strategies to different psychopathologies [measured with the Child Behaviour Checklist, CBCL, 37], they found specific strategies for affective and externalising disorders, but not for ADs.

Hence, so far, there is no questionnaire to measure avoidance and at the same time differentiate between various types of avoidance strategies, especially in dealing with anxiety symptomatology in children. The Bochum Assessment of Avoidance-based Emotion Regulation for Children (BAER-C) aims to fill this gap, following a top-down approach using the Process Model of Emotion Regulation [18,38]. The goal of the present study is to introduce this newly developed questionnaire and to examine its psychometric properties in a clinical CAD sample, and in a school sample. We hypothesize that the factor structure derived from the model will be found in the present data. We expect all the avoidance scales to form an avoidance factor, while the reappraisal scale stands on its own. The questionnaire is hypothesized to be internally consistent and to show good to excellent reliability. All avoidance strategies in the questionnaire are expected to correlate positively with anxiety symptoms, as well as with maladaptive emotion regulation proving convergent validity, however, not with externalizing symptoms showing divergent validity. Finally, children diagnosed with an anxiety disorder are expected to engage in more avoidance than the school sample, thus pointing to construct validity. To control for effects of gender and age in the development of ER [39], both variables will be included as fixed factor and covariate.

## 2 Methods

### 2.1 Development and Rational of the Bochum Assessment of Avoidance-based Emotion Regulation for Children (BAER-C)

The BAER-C was developed as a theory-driven 29-item instrument to assess different avoidance strategies as well as reappraisal in anxiety eliciting situations in children aged 8 to 18 years. The BAER-C was developed based on the process model of emotion regulation [18,38], following a top-down approach. Avoidance strategies covered by the BAER-C were matched to the ER stages situation selection, situation modification, and attentional deployment according to the Gross' model. Because of the lack of an avoidance strategy for the appraisal stage, the strategy "reappraisal" was added to the questionnaire. This complemented the

BAER-C with an adaptive ER strategy. Response modulation was excluded because the goal of the questionnaire was to focus on anticipatory, rather than responsive avoidance strategies. The items were constructed to assess the different avoidance strategies and reappraisal. Children answered on a five-point Likert scale ranging from 1 (totally disagree) to 5 (totally agree).

At the beginning of the questionnaire the stem statement "When I am anxious. . ." was presented. Every item started with the expression "I try" followed by the avoidance behaviour. This expression was chosen so that participants could answer the item regardless of the success of their ER. All items were formulated with easy wording in order to ensure children's understanding.

The first subscale is matched to the ER family situation selection and measures behavioural avoidance. This avoidance strategy is defined as avoiding the anxiety-eliciting object at all costs: Children may try to choose situations depending on the likelihood of encountering the feared object, that is, "When I am anxious, I try to avoid what makes me anxious". The second subscale depicts the ER family of situation modification. In contrast to situation selection, children enter the situation but try to change it to reduce their anxiety. A corresponding avoidance strategy for this is reassuring/security behaviour such as "When I am anxious, I try not to be alone". The next set of ER strategies refers to attentional deployment in which attention is focused on other things to suppress anxiety as captured in for example "When I am anxious, I try to think of nothing". The fourth subscale is derived from the appraisal family measuring reappraisal as an adaptive strategy: "When I am anxious, I try to make the best out of the situation". Thus, four subscales with 25 items were created. Following suggestions of a meta-analytic review of adaptive and maladaptive ER strategies in youth anxiety and depression [40] four items concerning the subjective skill to handle anxiety and the acceptance of anxiety were generated. The meta-analysis emphasizes that acceptance is associated with less avoidance, and that the feeling to regulate emotions in an adaptive way might also be connected to anxiety and depression in youths. One open question was added as a 30th item to encourage children to name an ER strategy not listed in the questionnaire. All but one item was worded without negations.

## 2.2 Procedure

To validate the BAER-C, two samples were collected in two different studies. In both studies the BAER-C was added to an already existing study protocol.

Study 1: To examine the questionnaire's factor structure and its relationship to anxiety disorders, the questionnaire was added to the ongoing PEACH study trial, which investigates the question whether parental involvement influences the treatment outcome in the psychotherapy of children with anxiety disorders. The BAER-C as well as other questionnaires were administered as part of a baseline assessment of the CBT treatment. The questionnaires were presented digitally on a computer. Children received support from trained assessors if they reported or showed difficulties with reading (especially if they were of a young age). The local Ethics Committee as well as the Ethics committee of the Deutsche Gesellschaft für Psychologie (DGPs) approved the study. The study was pre-registered at the German Clinical Trials Register (GermanCTR ID DRKS00009709).

Study 2: To analyse if the factor structure found in anxiety patients could be confirmed in a convenience sample and to examine construct validity by using another ER questionnaire, school children were assessed by undergraduate psychology students conducting a questionnaire validation study in German schools. The goal of the study was to validate several anxiety related questionnaires. The BAER-C was added to the questionnaire package presented as a paper-pencil version in class. Assistance by a researcher was offered when needed.

**Table 1. Sample characteristics.**

|  | Study 1: CAD Sample (n = 199) | Study 2: School Sample (n = 129) |
|---|---|---|
|  | M (SD) | M (SD) |
| Age | 10.97 (2.26) | 11.78 (1.17) |
| SCAS-C Total Score | 27.51 (14.20) | 23.55 (14.64) |
| SDQ Internalizing | 6.99 (2.03) | Not included |
| SDQ Externalizing | 13.20 (3.58) | Not included |
| FEEL-KJ Adaptive | Not included | 46.26 (13.02) |
| FEEL-KJ Maladaptive | Not included | 23.31 (8.40) |

BAER-C: Bochum Assessment of Avoidance-based Emotion Regulation for Children; SCAS: Spence Children Anxiety Scale; SDQ: Strength and Difficulties Questionnaire; FEEL-KJ: Fragenbogen zur Erfassung von Emotionsregulation bei Kindern und Jugendlichen.

Participating classes received a certificate as well as some sweets to be shared in class as an incentive. Children filled out the questionnaire only after written informed consent by the child and the legal guardians were given. The study was approved by the local Ethics Committee.

Table 1 gives an overview of the sample characteristics and the questionnaires used in the studies.

## 2.3 Participants

The total sample consisted of 328 children and adolescents. The school sample consisted of 129 children (age $M$ = 11.68 years; $SD$ = 0.99; range 8–14; 48.8% ♀). The clinical sample included 199 children with a primary DSM-5 anxiety disorder diagnosis (age $M$ = 10.78; $SD$ = 2.25; range 8–16; 62.1% ♀). The clinical group was diagnosed using the Diagnostic Interview for Mental Disorders in Children and Adolescents [Kinder-DIPS-OA, 34,41] with either a primary diagnosis of SepAD ($n$ = 78), SAD ($n$ = 49) or specific phobia ($n$ = 63). 67% had one comorbid diagnosis, 27% had two comorbid diagnoses, and 18% three or more comorbidities. Comorbid diagnoses were other anxiety disorders (76%), externalizing disorders (9%), depressive disorders (4%), and other disorders (Enuresis/Encopresis, OCD, Tic disorders, sleeping disorders; 11%).

## 2.4 Comparison of samples

The comparison of the group with anxiety disorders and the school sample showed that they did not differ significantly in gender, $\chi^2$ = 2.81, $p$ = .09, but the school sample was significantly older than the CAD group, $t$ (470) = -6.10, $p$ < .001. To control for gender and age in all analyses, both variables were included as covariates.

To compare anxiety symptoms between the two groups, two Bonferroni-corrected one-way ANCOVAs were calculated for the SCAS questionnaire (child and parent version) with age as a covariate and gender and group as fixed factors. In the child version, the anxiety group ($M$ = 27.49; $SD$ = 14.23) showed significantly higher levels of total anxiety symptoms, $F(1,315)$ = 4.52, $p$ = .03 $\eta_p^2$ = .01, than the student sample ($M$ = 23.55; $SD$ = 14.64). In addition, the

main effect of gender, $F(1,315) = 16.54$, $p < .001$, $\eta_p^2 = .05$, and the interaction between group-
*gender, $F(1,315) = 5,52$, $p = .019$, $\eta_p^2 = .02$, were significant. Pairwise comparisons showed
that girls did not differ significantly in levels of anxiety (CAD group $M = 28.58$, $SD = 14.69$;
student sample $M = 28.85$, $SD = 15.16$; $t(194) = -1.37$, $p = .17$), while boys in the CAD group
($M = 25.72$, $SD = 13.26$) showed a significantly higher SCAS-C score than boys in the student
sample ($M = 18.33$, $SD = 12.11$; $t(123) = -4.29$, $p < .001$). Seven percent of the children in the
school sample scored above the clinical cut-off.

For the parent version, the ANCOVA only showed a significant effect of group, $F(1,265) = 44.10$, $p < .001$, $\eta_p^2 = .14$, with children in the CAD group ($M = 31.04$; $SD = 12.78$) having sig-
nificantly higher scores than children in the student sample ($M = 19.46$, $SD = 12.10$). To com-
plement this data, severity rating of the Kinder-DIPS-OA structured interview was analyzed.
The mean severity rating was $M = 5.77$ ($SD = .82$) on a scale from 0 to 8 with no differences
found in relation to the gender of the children, $t(304) = .48$, $p = .63$, indicating that the clinical
sample used for this analyses suffered from moderate to severe anxiety disorders and is compa-
rable with severity ratings reported in treatment studies and meta-analyses [42,43].

## 2.5 Measures

### 2.5.1 Diagnostic Interview for Mental Disorders in Children and Adolescents–Open Access (Kinder-DIPS-OA).
All children in the CAD group were diagnosed with the Kinder-
DIPS-OA [41] by assessors who were psychotherapists for children and adolescents (either in
training or licensed) and were systemically trained and certified to conduct the interview. The
Kinder-DIPS-OA is a well validated structured interview to assess DSM-5 mental disorders in
children and adolescents in a classificatory and dimensional approach that has different forms
for the child and the parent [for an overview of the psychometric properties, see 34,44]. Diag-
noses were based on composite information from both the child and parent interviews.

### 2.5.2 Spence Children's Anxiety Scale (SCAS-C).
The SCAS-C [45] is a widely used
instrument to assess anxiety symptoms. The questionnaire consists of 44 items (38 anxiety
related, 6 positive filler items), which are rated on a scale from never (0) to always (3). The
scale measures six domains of anxiety (i.e., separation anxiety, social phobia, obsessive-com-
pulsive disorder, panic/agoraphobia, generalized anxiety), and the items can be summed up to
calculate a total anxiety score. The scale shows good to excellent psychometric properties
across several countries [for an overview, see 45]. A recent analysis with 1438 clinically diag-
nosed children showed good to excellent internal consistency (Cronbach's $\alpha = .88$ - .91) as well
as the ability to accurately identify children with separation anxiety and social anxiety disorder
[46]. In this study, we used the German version of the SCAS-C [47]. In its validation study, the
SCAS-C showed very good internal consistency (Cronbach's $\alpha = .92$) and split-half reliability
($r = .90$). The SCAS-C also showed convergent validity, correlating significantly with other
anxiety questionnaires as well as overall measures of childhood psychopathology [47]. In the
current sample, internal consistency of the anxiety scale was $\alpha = .86$.

### 2.5.3 Strength and Difficulties Questionnaire (SDQ).
The SDQ is an internationally
established screening questionnaire for children aged 3 to 16 years to measure overarching
psychopathology, consisting of 25 items with five subscales (emotional symptoms, conduct
problems, hyperactivity/inattention, peer relationship problems, prosocial behaviour). Each
item is rated on scale from not true (0) to certainly true (2). It is possible to calculate an inter-
nalizing (emotional and peer problems), an externalizing score (conduct and hyperactivity)
and an overall score (all four difficulties scales). The SDQ exists in a child and parent version
and shows good psychometric properties [Cronbach's $\alpha = .73$; see 48]. Goodman, Ford, Sim-
mons, Gatward and Meltzer [49] demonstrated the SDQ's ability to correctly screen for mental

disorders. The study showed that the questionnaire was especially effective to screen for ADHD, conduct disorder, and depressive disorder, but only moderately suited to screen for anxiety disorders. Thus, in our study the SDQ was used to calculate discriminant validity using the externalizing subscale. The German self-report version [50], which was used in this study, showed low to good measures of reliability (Cronbach's $\alpha$ = .55 - .77). In the current sample, internal consistency of the internalising scale was $\alpha$ = .63 and $\alpha$ = .66 for the externalizing scale.

**2.5.4 Questionnaire to Measure Emotion Regulation in Children and Adolescents (Fragebogen zur Erhebung der Emotionsregulation bei Kindern und Jugendlichen, FEEL-KJ).** The German ER questionnaire FEEL-KJ [35] measures 15 different ER strategies using two items per strategy. The questionnaire has a two-factor structure (adaptive vs. maladaptive strategies) and assesses the use of the ER strategies for anxiety, anger and sadness. Adaptive strategies are problem solving, acceptance, forgetting, distraction, humour enhancement, revaluation and cognitive problem solving. Maladaptive strategies include giving up, withdrawal, rumination, self-devaluation, and aggressive actions. In this study, only the anxiety version was used. The secondary scales, adaptive and maladaptive strategies, show good to excellent psychometric properties [Cronbach's $\alpha$ = .82 - .92; see 35]. Most research with the FEEL-KJ has been conducted using the Dutch version of the questionnaire [51]. The two-factor structure was confirmed, however, authors tentatively suggested that the underlying structure might be even more complex. In this study, the secondary scales showed good internal consistencies $\alpha$ = .82 for maladaptive and $\alpha$ = .90 for adaptive ER strategies.

## 2.6 Statistical analysis

To analyse reliability, and the factor structure of the BAER-C, data from the clinical sample was used. Item analysis included analysis of item difficulty, skewness, discriminatory power and internal consistency (Cronbach's $\alpha$) as a measure of reliability. In addition, McDonald's $\omega$ was used to assess model-based reliability. To investigate the factor structures, exploratory factor analysis was conducted, choosing maximum-likelihood estimation as well as a promax rotation considering the interdependence of the different ER strategies. All factor loadings higher than .32 were considered acceptable [52]. To test construct, convergent and discriminant validity, Spearman correlations were calculated.

Data from the student sample was used to check the factor structure found in the clinical sample with confirmatory factor analysis using robust maximum likelihood estimation. This was added to the analysis to determine whether there is statistical evidence of a second order avoidance score, consisting of all avoidance subscales. In addition, the FEEL-KJ ER questionnaire was used to further explore construct validity.

A multivariate analysis of covariance (MANCOVA) with age as covariate and gender as fixed factor was conducted to analyse differences between the CAD group and the school sample and to check for construct validity. Bonferroni corrected post-hoc tests were used to analyse subgroups. Analyses were performed with IBM SPSS Statistics 24 [53], as well as RStudio [54] with the packages lavaan [55] and psych [56] for factor analyses.

## 3 Results

### 3.1 Item and factor-analysis in a clinical sample

Analysis of item difficulty and discriminatory power led to the exclusion of three items. These items were situational items (e.g., trying to hide in the room) and were removed because of a low discriminatory power ($<$ .30).

**Table 2. Results of the exploratory factor analysis, maximum likelihood estimation, promax rotation.**

| | Factor | | | | |
|---|---|---|---|---|---|
| | 1 Reappraisal | 2 Behavioural Avoidance | 3 Social Reassurance | 4 Suppression | 5 Verbal Reassurance |
| Item 15 | .97 | | | | |
| Item 21 | .91 | | | | |
| Item 27 | .65 | | | | |
| Item 19 | .64 | | | | |
| Item 23 | .52 | | | | |
| Item 4 | .49 | | | | |
| Item 16 | .40 | | | | |
| Item 29 | *.27* | | | | |
| Item 20 | | .77 | | | |
| Item 13 | | .75 | | | |
| Item 26 | | .68 | | | |
| Item 9 | | .57 | | | .30 |
| Item 3 | | .56 | | | |
| Item 2 | | .53 | | | |
| Item 10 | *.20* | | | | |
| Item 18 | | | .90 | | |
| Item 22 | | | .87 | | |
| Item 12 | | | .76 | | |
| Item 5 | | | | .86 | |
| Item 17 | | | | .76 | |
| Item 25 | | | | | *.25* |
| Item 1 | | | | *.23* | |
| Item 7 | | | | | .93 |
| Item 8 | | | | | .68 |
| Item 28 | | | | | .33 |

For a description of the items, see Table 3. Cutting value to extract factors was set to .32 [52].

To explore the questionnaire's factor structure, factor analysis was conducted with all the remaining items. Parallel analysis as well as a scree plot analysis recommended a five-factor structure (see S1 Fig and S2 Table for data and simulated eigenvalues). The five-factor structure was chosen due to the results of the parallel-analysis. Although a three-factor solution (using the Kaiser-Guttmann criterion) or even a one-factor solution (interpreting the huge drop of the eigenvalues between the first and second factor) would have been possible as well, the five-factor variant was chosen to not only match the theoretical foundation, but to maximize the explained variance. This is supported by the fact that the items, that were added in an exploratory approach and did not directly fit into the model, were removed in the item analysis.

Therefore, a factor analysis with maximum likelihood extraction as well as a nonorthogonal promax rotation was conducted. Table 2 shows the results of the five-factor model explaining 54% of the variance with the new factor structure.

Factor 1 was comprised of seven items with factor loadings of .40 to .97 and can be interpreted as the reappraisal factor of the Gross' model. Factor 2 was comprised of six items with factor loadings from .53 to .77. Factor 2 can be interpreted as the behavioural avoidance/

situation selection factor. The third and the fifth factor were comprised of the items assumed to belong to the situation selection scale. Three items loaded on Factor 3 with factor loadings of .76 to .90, and two items loaded on Factor 5 with loadings of .86 and .76. Both factors can be interpreted as factors of the situation modification family. Factor 3 focuses on social reassurance to reduce anxiety; Factor 5 describes verbal reassurance behaviour. Factor 4 is comprised of three items with factor loadings of .33 to .93. The factor can be interpreted as the suppression factor Item 16 ("I try to tell myself, that it is not so bad") loaded both on the reappraisal (.40) and the suppression (.47) factor. Due to better conceptual fit, the item was sorted to the reappraisal scale. Items 1 and 25, as well as the subjective ER items (6, 10) and acceptance item (29) failed to load sufficiently on any factor and thus were removed. The factors showed medium to high inter-correlations ranging from .31 to .67 with generally high correlations between the four avoidance scales. For a detailed overview, see S1 Table in the electronic supplements. A TLI of .873 and a RMSEA index of 0.072 indicated an acceptable model fit.

Following the results of the factor analysis, it is possible to calculate mean scores for the five subscales, as well as an overarching avoidance score, which is calculated by taking the overall mean of the behavioural avoidance, social reassurance, verbal reassurance and suppression subscales.

### 3.2 Reliability (internal consistency) and skewness

The new 21-item, five-factor questionnaire showed very good internal consistency with McDonald's $\omega$ = .94. Internal consistency of the five subscales ranged from .78 to .85 (Table 3). The internal consistency of the avoidance scale was $\alpha$ = .86. Analysis of skewness showed most items to be moderately skewed, with some items on the behavioural avoidance scale showing strong right-sided skewness (Table 3).

### 3.3 Convergent validity in a CAD sample

To further analyse the validity of the BAER-C, Spearman correlations were calculated correlating the BAER-C scores with other psychometric measures. All avoidance subscales, except suppression, indicated significant correlations with the anxiety symptom scale of the SCAS-C (all $rs > .17$, all $ps < .05$, Table 4).

In addition, the BAER-C avoidance score showed a significant correlation with the total anxiety symptom scale of the SCAS-C ($r = .23$, $p < .001$) therefore indicating convergent validity. Surprisingly only the social reassurance subscale correlated significantly with the internalizing symptoms subscale of the SDQ.

### 3.4 Divergent validity in a CAD sample

Contrary to the hypothesis, the avoidance scales suppression and reappraisal correlated significantly with the externalizing subscale of the SDQ (all $r$ from .12 to .35, all $p < .01$, see Table 4 for details). In contrast, behavioural avoidance, social reassurance, and verbal reassurance did not correlate significantly with externalizing symptoms, therefore indicating discriminant validity for these subscales ($r = .01$, $p = .85$).

### 3.5 Confirmation of the factor structure in a student sample

To confirm the factor structure found in the clinical sample, confirmatory factor analysis was done in the community sample. The model was built using the scale structure described in Table 3. In addition, a second order factor for the avoidance score, consisting of all avoidance subscales, was added to the model. Results showed acceptable model fit. Although the $\chi^2$ test

**Table 3. Overview of the 21-item, five-factor version of the Bochum Assessment of Avoidance-based Emotion Regulation for Children (BAER-C).**

|  | Number of Items | Skew | Cronbach's $\alpha$ |
|---|---|---|---|
| **Behavioural Avoidance** | 6 |  | .81 |
| Item 2. . . I try to avoid what makes me anxious. |  | -1.31 |  |
| Item 3. . . I try to withdraw from the situation. |  | -.99 |  |
| Item 9. . . I try to get out of the way of what makes me afraid. |  | -1.33 |  |
| Item 13. . . I try to avoid the situation in the future. |  | -.80 |  |
| Item 20. . . I try not to get in the situation in the future. |  | -.86 |  |
| Item 26. . . I try everything I can so that I do not encounter what makes me afraid. |  | -.64 |  |
| **Verbal Reinsurance** | 2 |  | .79 |
| Item 5. . . I try to ask again and again if everything is going to be okay. |  | .04 |  |
| Item 17. . . I try to ask someone if everything is okay. |  | .36 |  |
| **Social Reassurance** | 3 |  | .88 |
| Item 12. . . I try not to be alone. |  | -.81 |  |
| Item 18. . . I try to have someone with me. |  | -.91 |  |
| Item 22. . . I try to have another person with me. |  | -.88 |  |
| **Suppression** | 3 |  | .78 |
| Item 7. . . I try to push what makes me afraid out of my head. |  | -.89 |  |
| Item 8. . . I try to forget what makes me afraid. |  | -.90 |  |
| Item 28. . . I try not to think about what makes me afraid. |  | -.85 |  |
| **Avoidance Scale** | 14 |  | .86 |
| **Reappraisal** | 7 |  | .88 |
| Item 4. . . I try to think more positive about the situation. |  | -.37 |  |
| Item 15. . .I try to make the best out of the situation. |  | -.35 |  |
| Item 16. . .I try to tell myself that it is not that bad. |  | -.63 |  |
| Item 19. . .I try to tell myself that I can do it. |  | -.60 |  |
| Item 21. . .I try to find something good in the situation. |  | -.03 |  |
| Item 23. . .I try to ignore what makes me afraid. |  | -.56 |  |
| Item 27. . .I try to find explanations for the situation that reduce my fear. |  | -.14 |  |
| **Total** | 21 |  |  |

showed significance ($\chi^2$(184, 119) = 288.81, p < .00), the ratio between the chi square statistic and the degrees of freedom showed an acceptable model fit $\chi^2/df = 1.54 < 2$ [57,58]. RSMEA = .07 and SRMR = .08 as well as a CFI and TLI were slightly below the threshold of 0.95 (CFI = .91, TLI = .89) and also suggest an acceptable model fit. Standardized path estimates are shown in Fig 1.

### 3.6 Convergent and divergent validity with the SCAS and FEEL-KJ in a student sample

To confirm the convergent validity of the BAER-C, the correlation analysis with the SCAS-C scale was repeated in the student sample. Here, all avoidance scales, as well as the reappraisal scale, show significant correlations with the total anxiety scale of the SCAS-C (Table 5).

To analyse the relation of the BAER-C and its subscales with other ER questionnaires, correlations with the FEEL-KJ questionnaire's second-order factors were calculated. Confirming the hypothesis, all avoidance scales correlated with the maladaptive factor of the FEEL-KJ (Table 5). Contrary to the hypothesis, all avoidance scales except Social Reassurance correlated

Table 4. Correlations of the BAER-C subscales and the avoidance score with the anxiety scale of the SCAS-C.

| | 1<br>Behavioural Avoidance | 2<br>Verbal Reinsurance | 3<br>Social Reassurance | 4<br>Suppression | 5<br>Reappraisal | 6<br>Avoidance Score |
|---|---|---|---|---|---|---|
| SCAS-C<br>Total Score | .17* | .31** | .21** | .03 | -.06 | .23** |
| SDQ<br>Internalizing | .03 | .06 | -.16* | .05 | -.08 | -.01 |
| SDQ<br>Externalizing | .02 | .08 | .12 | .23* | .35** | .12 |

SCAS-C: Spence Children Anxiety Scale, SDQ: Strength and Difficulties Questionnaire

$^*$ $p < .05$

$^{**}$ $p < .01$.

with the adaptive strategies factor. Therefore, the expected divergent validity of the questionnaire could not be shown.

## 3.7 Differences in the BAER-C subscales

To analyse differences in the BAER-C subscales between both samples, a MANCOVA using group and gender as fixed factors and age as a covariate was calculated. The MANCOVA showed a significant main effect for group, $F(5,304) = 12.63$, $p = .003$, $\eta_p^2 = .17$, when corrected for age. It also showed a significant main effect for gender, $F(5,304) = 2.60$, $p < .05$, $\eta_p^2 = .04$, and age, $F(5,304) = 3.94$, $p < .01$, $\eta_p^2 = .06$. No significant interaction effect was found.

Further Bonferroni-corrected post-hoc analyses showed that the effect for group was found in behavioural avoidance, $F(1,305) = 21.59$, $p < .001$, verbal reassurance, $F(1,305) = 4.1$, $p = .04$, and social support $F(1,305) = 31.79$, $p < .001$, but not in suppression and reappraisal (Table 6).

In a Bonferroni-corrected post-hoc analysis for the gender effect, only verbal reassurance differed significantly between genders, $F(1,305) = 7.05$, $p = .008$, with girls showing higher use of this strategy than boys ($M_{girls} = 3.93$, $SD_{girls} = 1.12$; $M_{boys} = 3.51$, $SD_{boys} = 1.10$).

## 4 Discussion

The aim of the current study was to introduce the newly developed BAER-C questionnaire to measure various avoidance strategies and reappraisal in anxious situations and to present first data on its psychometric evaluation in two different child samples. The BAER-C has a strong theoretical foundation and shows good usability and acceptance. The hypothesized four-factor model was discarded in favour of a five-factor model that explains a satisfactory amount of variance and can be interpreted in line with the process model of ER [18,38].

The Behavioural Avoidance, Suppression, and Reappraisal factors were included in the new model, although, Item 23 ("...I try to ignore what makes me afraid") shifted from suppression to reappraisal. The children seem to have interpreted this item in a different way than the developer of the scale, possibly thinking of something positive when ignoring what makes them afraid. This shows the importance of validating newly developed questionnaires in the targeted group. Nonetheless, the item is formulated to measure suppression. Thus, it should be carefully investigated in a follow-up study. The factor Security Behaviour splits into two highly correlated factors both describing strategies to alter the situation to reduce anxiety. The first factor focuses on verbal strategies like talking to another person (Verbal Reassurance). The second factor highlights social reassurance, for example, the presence of another person to

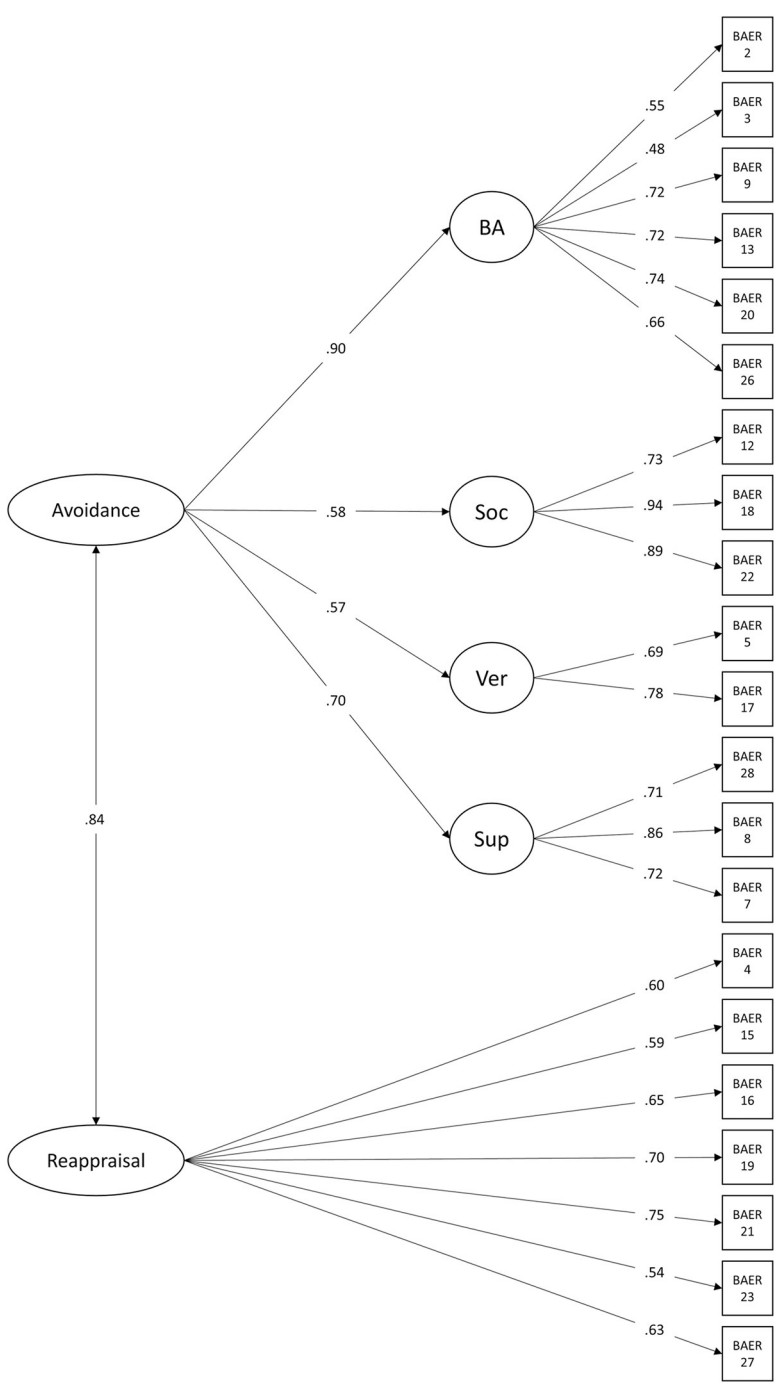

**Fig 1. Results of the second order confirmatory factor analysis.** All arrows show significant associations, p < .001; BA: Behavioural Avoidance, Soc: Social Reassurance, Ver: Verbal Reassurance, Sup: Suppression.

reduce fear (Social Reassurance). It seems the factor split up in a more active variant of modifying behaviour (talking to someone), in contrast to a more passive approach (trying that someone is there with the child). Therefore, the new factor structure enables the BAER-C to measure a total of four different avoidance strategies, as well as reappraisal, while maintaining its strong theoretical foundation. The factor structure was confirmed via a CFA in the second

**Table 5. Correlations of the BAER-C subscales and the avoidance score with the anxiety scale of the SCAS-C and the adaptive and maladaptive scales of the FEEL-KJ.**

| | 1 Behavioural Avoidance | 2 Verbal Reinsurance | 3 Social Reassurance | 4 Suppression | 5 Reappraisal | 6 Avoidance Score |
|---|---|---|---|---|---|---|
| SCAS-C Total Score | .33** | .43** | .35** | .20* | .21* | .25** |
| FEEL-KJ Adaptive | .29* | .28** | .13 | .35** | .52*** | .33** |
| FEEL-KJ Maladaptive | .32** | .41** | .28** | .24** | .32** | .38** |

SCAS-C: Spence Children Anxiety Scale, FEEL-KJ: Fragebogen zur Erhebung der Emotionsregulation bei Kindern und Jugendliche

$^*$ $p < .0$

$^{**}$ $p < .01$.

sample. Furthermore, the new factor structure showed good to very good internal consistency, comparable with other questionnaires measuring avoidance [31]. Analysis of skewness showed some items to be strongly skewed, especially on the behavioural avoidance scale. Although some right-sided skew was expected because of the use of a sample of children with an anxiety disorder, such a high level of skewness might have led to the underestimation of the models. This problem should be addressed in future studies with a more balanced sample consisting of children with and without anxiety disorders.

All avoidance scales (except suppression), as well as the avoidance score, showed small to moderate associations with anxiety symptoms and maladaptive ER, thus partly confirming convergent validity, especially with respect to the important association between anxiety and avoidance. This is, once again, comparable with the results of the CAMS questionnaire [31]. Nonetheless, the questionnaire's scales failed to correlate with the internalising subscales of the SDQ, which could partly be explained by the fact that the SDQ shows weaknesses in screening for anxiety disorders [49]. Correlations could also be attenuated because of the rather low reliability of the SDQ in this sample. Thus, the high correlation of avoidance measured with the SCAS-C questionnaire seems to be a more reliable source to investigate construct validity. Anyhow, future studies should use other screening instruments to explore whether the BAER-C correlates with internalizing symptoms.

Another unexpected result contradicting the hypothesis was the significant correlation of the avoidance scales and adaptive ER strategies in the FEEL-KJ questionnaire. This could point to an adaptive function of avoidance in a certain situation, as proposed by Hofmann and Hay [22]. Nonetheless, this result limits the construct validity of the questionnaire. A follow-up study should carefully examine the BAER-C subscales with respect to other ER

**Table 6. Post-hoc comparison of the children with anxiety disorder and student sample.**

| | CAD sample M (SD) | Student Sample M (SD) | F | Df | p | Effect size $\eta_p^2$ |
|---|---|---|---|---|---|---|
| Behavioural Avoidance | 4.12 (.77) | 3.66 (.87) | 21.59 | 1,305 | < .001 | .06 |
| Verbal Reassurance | 3.20 (1.00) | 2.97 (.93) | 4.10 | 1,305 | .04 | .01 |
| Social Reassurance | 4.01 (1.07) | 3.31 (1.09) | 31.79 | 1,305 | < .001 | .09 |
| Suppression | 4.06 (.94) | 4.08 (.97) | 0.23 | 1,305 | .63 | .00 |
| Reappraisal | 3.65 (.90) | 3.79 (.84) | 1.75 | 1,305 | .19 | .00 |

Means corrected for age and Bonferroni-correction for multiple comparisons.

questionnaires, although such German language instruments are rather scarce. In addition, questionnaire data could be correlated with behavioural measures of avoidance in a lab situation or in daily life monitoring.

Further analysis of divergent validity of behavioral avoidance, social reassurance and verbal reassurance showed no correlations with externalizing symptoms. In contrast, the strategies suppression and reappraisal showed small associations with externalizing symptoms in the SDQ. These results are in line with anxiety questionnaires [47] which show moderate to large associations with externalizing symptoms. However, the findings could also support the idea that avoidance as an ER strategy might be more complex and not only connected to anxiety but also other psychological disorders; pointing to the role of ER as a transdiagnostic factor [14]. Certainly, there is need for more research with data that allows the comparison of clinically diagnosed children with internalizing and externalizing disorders.

Compared with the school sample, the questionnaire also showed that children with ADs more frequently engage in avoidance behaviour. Nonetheless, these results can only be applied to the strategies behavioural avoidance, verbal reassurance, and social support. For these sub-scales, the BAER-C is in line with questionnaires measuring avoidance [31] as well as with prominent anxiety models [59] showing an association between ADs and avoidance as an ER strategy. In addition, our analysis showed a significant effect of gender in social support. This is in line with previous findings that girls in general and independent of their anxiety status use more ER strategies involving social factors than boys, especially with respect to internalizing emotions like sadness or anxiety [39,60].

The results with respect to the cognitive avoidance strategies suppression and reappraisal show no differences between CAD and the student sample. This is rather surprising considering studies in which CADs showed more suppression and a lower rate in the use of reappraisal [40,61,62]. This could indicate that these two scales warrant clarification.

Interestingly, the data of the child version of the SCAS-C point to a low level of anxiety symptoms in both groups. In contrast, the severity rating of the Kinder-DIPS-OA confirms that the CAD sample is indeed a clinical sample with moderate to severe diagnoses.

## Limitations

Although the results provide first evidence for the factor structure and the reliability of the BAER-C, validity can only be confirmed for some parts of the instrument. Some problems with validity might be due to limitations of the study design. As the questionnaire was added to already existing projects, validation questionnaires used in the two projects differed. This resulted in differences in sample sizes in the analyses. In addition, including additional ER questionnaires to sufficiently examine construct validity was not possible. Furthermore, it was not possible to analyse test-retest-reliability or test sensitivity to change (e.g., in a CBT-treatment for CAD), which should be subject of future studies.

Nonetheless, the BAER-C is the only theory-based instrument so far to measure various avoidance strategies in anxiety-inducing situations. Regarding factor structure and reliability, this first attempt to examine the questionnaire was successful. In particular the scales behavioural avoidance, verbal reassurance and social support show promising results in psychometrics as well as in differentiating between CADs and a school sample, which speaks for the generalizability of the scales. In contrast, the results concerning validation of the scales suppression and reappraisal require more research or even a rework in order to accurately assess their strategy. Further clarification is needed regarding the significant correlation with adaptive ER strategies investigating the relationship of the BAER-C with other ER questionnaires. This relation must be explored in detail to understand whether the BAER-C has structural

issues or if the role of avoidance is as diverse as Hofmann and Hay [22] propose. To further investigate generalizability, future studies should include older and younger samples, as well as children with other mental disorders.

Besides the possibilities to use the BAER-C in research, the questionnaire could also deepen the understanding of patients' avoidance strategies in treatment. The questionnaire can help to analyse avoidance in CADs on behavioural, social, and cognitive levels, thus adding important information to form an individualized model of the disorder for the patient as well as the therapist. The more detailed the picture of a patient's avoidance behaviour, the easier it is for therapists to develop strategies and exposure tasks to reduce avoidance as a maintaining factor of the AD. In addition, exposure therapy can become more effective when targeting specific avoidance behaviours.

## 5 Conclusion

In summary, this study introduces the BAER-C as a promising instrument with a strong theoretical foundation focusing on different avoidance strategies in the context of CADs. Results of the factor structure and reliability are promising, while results on validity are ambiguous and require more research. The questionnaire has many potential applications in research as well as in clinical practice to improve the understanding of avoidance as an ER strategy in the elicitation and maintenance of CADs. Further research is needed to clarify validity. Nonetheless, the BAER-C can be expected to be a valuable addition to existing questionnaires in the context of CADs.

## Supporting information

**S1 Fig. Scree plot of the parallel-analysis suggesting a five-factor solution.**
(DOCX)

**S1 Table. Inter-factor correlations of the BEAR-C subscales.**
(DOCX)

**S2 Table. Additional results of the factor analysis as well as simulated eigenvalues of the parallel analysis.**
(DOCX)

## Acknowledgments

We thank all the families, children and therapists who were part of the KibA-study, as well as Helen Copeland-Vollrath and Christine Katzmann for proof-reading.

## Author Contributions

**Conceptualization:** Jan Schomberg, Silvia Schneider.

**Data curation:** Michael W. Lippert, Jan Schomberg.

**Formal analysis:** Michael W. Lippert, Katharina Sommer, Tabea Flasinski.

**Funding acquisition:** Silvia Schneider.

**Investigation:** Michael W. Lippert, Jan Schomberg.

**Methodology:** Michael W. Lippert, Verena Pflug, Silvia Schneider.

**Project administration:** Michael W. Lippert, Verena Pflug, Hanna Christiansen, Tina In-Albon, Susanne Knappe, Marcel Romanos, Brunna Tuschen-Caffier, Silvia Schneider.

**Resources:** Michael W. Lippert, Silvia Schneider.

**Supervision:** Verena Pflug, Tina In-Albon, Silvia Schneider.

**Validation:** Michael W. Lippert.

**Writing – original draft:** Michael W. Lippert, Katharina Sommer, Tabea Flasinski.

**Writing – review & editing:** Michael W. Lippert, Katharina Sommer, Tabea Flasinski, Verena Pflug, Hanna Christiansen, Tina In-Albon, Susanne Knappe, Brunna Tuschen-Caffier, Silvia Schneider.

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
