## [Decision Letter · Decision Letter 0]

8 Nov 2021

PONE-D-21-23200Measuring Avoidance and Emotion Regulation in Children – Development and Evaluation of the Bochum Avoidance and Emotion Regulation Scale for Children (BAER-C)PLOS ONE

Dear Dr. Lippert,

Thank you for submitting your manuscript to PLOS ONE. After careful consideration, we feel that it has merit but does not fully meet PLOS ONE’s publication criteria as it currently stands. Therefore, we invite you to submit a revised version of the manuscript that addresses the points raised during the review process.

I thank the authors for their patience and the reviewer for their comments on the manuscript. I was hoping to receive a second set of comments, but I was unable to secure an additional review. I have also reviewed the manuscript myself. I see that there are some strong aspects of the work. There is a clear motivation for the work and connections with models of emotion regulation. Here, you also extend those models to be applicable to children. The reviewer is very clear and pointed in their identification of several conceptual and analytic issues in the work. Their comments are comprehensive, so I will not reiterate them. I want to be clear that all will need to be addressed for a revision to be successful. 

I also have some queries about the measure. I did not see a description of the number of response options or the degree of skew in the indicators. If the items are rather skewed, that could explain some of the poor model fit. Relatedly, there are estimation options available that provide model fit information from EFAs (including within lavaan). Thus, the transition from CFA to EFA does not fully remove the need to report complete model fit information. 

We look forward to receiving your revised manuscript.

Kind regards,

Thomas M. Olino

Academic Editor

PLOS ONE

Journal Requirements:

2. Please include a copy of the scale developed in this study, in both the original language and English, as Supporting Information.

Reviewers' comments:

Reviewer's Responses to Questions

**Comments to the Author**

1. Is the manuscript technically sound, and do the data support the conclusions?

Reviewer #1: Partly

2. Has the statistical analysis been performed appropriately and rigorously? 

Reviewer #1: Yes

3. Have the authors made all data underlying the findings in their manuscript fully available?

Reviewer #1: Yes

4. Is the manuscript presented in an intelligible fashion and written in standard English?

Reviewer #1: Yes

5. Review Comments to the Author

Reviewer #1: The present study represents an important potential advance in theoretically driven measurement instruments for emotion regulation, and the instrument shows promise. However, the introduction would benefit from more clarity about the narrow focus of the measure, and more detail about the EFA and CFA need to be provided in the results. The most significant concerns are around the combination of two separate samples and the lack of validity evidence for the measure. Specific feedback follows.

Introduction

• It would be helpful to include examples of behavioral avoidance, safety behavior, and thought suppression when they are mentioned in the introduction

• The title of the questionnaire is confusing because it implies avoidance is separate from emotion regulation but the questionnaire was developed along the theory that avoidance is part of the process of emotion regulation. I appreciate the goal of the measure though, to develop a more appropriate assessment of the Gross process model, as my opinion is that the current Gross questionnaire (focusing on reappraisal and suppression) does not adequately assess this model! This has always bothered me, so I really appreciate these authors’ approach. That said it is a specific aspect being tested here, avoidance as it pertains to the process model, so however the title of the questionnaire could potentially capture that would be best. I’m not sure why avoidance strategies as they pertain to response modulation were excluded, but again, as long as it’s made clear throughout that this measure pertains to anticipatory strategies only, it’s not necessarily an issue. Also, to narrow it down even further, the questionnaire just assesses regulation of anxiety specifically! It’s going to be tough to capture this very specific narrow approach in the title of the measure so I’m not sure how to address all this. “Avoidance-based anticipatory emotion regulation strategies for anxiety scale for children” is more than a little clumsy! Maybe a short title for the measure but with an explanatory clause in the title of the article would help (e.g., ‘______’: A measure of ______ for children).

Method

• It’s unclear why these extra items were generated “In addition, items concerning the subjective skill to handle anxiety and the acceptance of anxiety were generated and all but one were worded without negations” as well as the extra 30th item. Were these items included in the analyses? If so, which items were they?

• It’s strange that the CFA was just mentioned in passing in the method section rather than given a full section in the results with all the appropriate information provided.

• The differences in administration and characteristics between the two samples are a little concerning. Ideally the samples would be analyzed separately and assessed for any differences between them, or at least this would be done and noted in a footnote if no differences were found.

• Were all participants given potential reading assistance or only in the second sample where it is noted?

• The confirmatory factor analysis is mentioned again in passing in the statistical analysis section, and described as unsuccessful—this is confusing, and once again, more detail should be provided about these analyses (at the very least, more of the fit statistics, and an indication of exactly what model was tested). A little more detail is provided in the results section but it’s unclear each time it’s mentioned previously that it will be discussed later in the paper and even when it comes up in the results section more information should be provided. It’s an unusual strategy to do the CFA first and then do an EFA when the CFA fails. Usually an EFA would be conducted first and the CFA to confirm the structure that was uncovered. Although the measure was developed based on theory, it still seems the traditional analytic process would make more sense. The CFA should also generally be done in a separate sample than the EFA. So, ultimately, it might make more sense to leave the CFA out of this paper entirely and conduct a follow up study to confirm the EFA structure in a new sample.

Results

• More information should be provided on the EA in the determining of the number of factors – numeric results from the parallel analysis, the scree plot, the Eigenvalues and % of variance explained, etc.

• Exactly how much would the overall reliability decline if item 1 was removed, because ideally items with low factor loadings should not be retained?

• It’s unclear whether item 9 was retained on both subscales.

• Item 23 does not seem to make sense on the reappraisal subscale and may need to be removed

• Although it states in text this information is in Table 2, I don’t see the factor intercorrelations in the table.

• The lack of correlations with the SDQ internalizing should be mentioned under the convergent validity header (as that was their intention), not divergent. Thus, convergent validity is only partially confirmed. Similarly, the correlations with adaptive subscale should be listed under divergent validity. There is a problem with the divergent validity evidence, as there is not evidence of divergent validity for this scale.

• It’s unclear why some analyses were done with the samples separated and some were not.

• It seems like it should say p < .001, not p > .001.

Discussion

• Unfortunately, I would not agree that the construct has been confirmed, given the poor divergent evidence. Ultimately though, the biggest issue is there are not enough validity measures in the study, so it’s difficult to draw conclusions. There should be validity measures that are more other existing emotion regulation and strategy questionnaires, not just validity measures of psychopathology. It’s one thing to show relations to externalizing problems where none were expected – as the authors state, ER is a transdiagnostic factor that is related to a lot of pathology – however, avoidance should be less related to externalizing, so it’s still concerning. However the correlations with adaptive coping are more concerning, given the lack of other relations to other ER measures among which to contextualize this.

• The authors should interpret why the security scale possibly ended up splitting into two separate factors.

• There should be a more thorough discussion of limitations.

6. PLOS authors have the option to publish the peer review history of their article (what does this mean?). If published, this will include your full peer review and any attached files.

Reviewer #1: No

---

## [Author Response · Author response to Decision Letter 0]

30 Mar 2022

Dear Prof. Olino, dear Reviewer, 

Thank you very much for this fair and thorough review. In my opinion your review helped a lot to improve the manuscript and therefore our questionnaire. I hope we have answered all your comments satisfyingly. Please be aware that, following the reviewer’s comments, we redid all analyses splitting the two samples, so the manuscript with tracked changes might be rather confusing. We look forward to hearing your opinion and your feedback on the revised manuscript and are thankful for your help in improving our work. 

Editor comments

I also have some queries about the measure. I did not see a description of the number of response options or the degree of skew in the indicators. If the items are rather skewed, that could explain some of the poor model fit. 

Answer: Thank you for noticing this. There is a description of the response scale in the methods sections (lines 162,163). We have added the results from the skewness (lines 344ff) analysis to the results section and added a part to the limitations regarding model fit (lines 468ff).

Relatedly, there are estimation options available that provide model fit information from EFAs (including within lavaan). Thus, the transition from CFA to EFA does not fully remove the need to report complete model fit information.

Answer: Thank you for this helpful comment. We have added model fit information for the EFA to the results section. (lines 338,339)

5. Review Comments to the Author

Reviewer #1: The present study represents an important potential advance in theoretically driven measurement instruments for emotion regulation, and the instrument shows promise. However, the introduction would benefit from more clarity about the narrow focus of the measure, and more detail about the EFA and CFA need to be provided in the results.

The most significant concerns are around the combination of two separate samples and the lack of validity evidence for the measure. Specific feedback follows.

Introduction

* It would be helpful to include examples of behavioral avoidance, safety behavior, and thought suppression when they are mentioned in the introduction

Answer: Thank you for this comment. I have included some examples in the introduction. (lines 90ff) 

* The title of the questionnaire is confusing because it implies avoidance is separate from emotion regulation but the questionnaire was developed along the theory that avoidance is part of the process of emotion regulation. I appreciate the goal of the measure though, to develop a more appropriate assessment of the Gross process model, as my opinion is that the current Gross questionnaire (focusing on reappraisal and suppression) does not adequately assess this model! This has always bothered me, so I really appreciate these authors' approach. That said it is a specific aspect being tested here, avoidance as it pertains to the process model, so however the title of the questionnaire could potentially capture that would be best. I'm not sure why avoidance strategies as they pertain to response modulation were excluded, but again, as long as it's made clear throughout that this measure pertains to anticipatory strategies only, it's not necessarily an issue. Also, to narrow it down even further, the questionnaire just assesses regulation of anxiety specifically! It's going to be tough to capture this very specific narrow approach in the title of the measure so I'm not sure how to address all this. "Avoidance-based anticipatory emotion regulation strategies for anxiety scale for children" is more than a little clumsy!

Maybe a short title for the measure but with an explanatory clause in the title of the article would help (e.g., '______': A measure of ______ for children).

Answer: Thank you for this thoughtful comment. After reconsidering all your argumentation, we agree that the original title might have been confusing. We have changed the title in accordance with your comment to “Bochum Assessment of Avoidance-based Emotion Regulation for Children (BAER-C):Development and Evaluation of a new instrument measuring anticipatory avoidance based emotion regulation in anxiety eliciting situations”. In this way we can keep our acronym while still emphasizing the actual content of the questionnaire and follow your suggestion. 

Method + results: 

Answer: Thank you for your thorough review and all your comments on the methods and results. We apologize for the confusion of having two samples integrated into one. We understand that having two samples in which not all participants filled out the same questionnaires is problematic and highly confusing. To solve this, we have decided to restructure our study and redo all calculations. To solve the sample problem, we have separated the samples. In a first step, we have used the CAD sample to do an exploratory factor analysis, like you suggested. We also did item analyses as well as an analysis of internal consistency and parts of construct validity using all the questionnaires we have in the CAD sample. In a second step, we used the student sample to confirm the factor structure found in the CAD sample with a CFA. We also used the FEEL-KJ data for additional information on construct validity. In a third step, we compared both samples for criterion validity. Because of the new factor analysis, which only used the CAD sample, two additional items were removed because of low factor loadings. That leads to different results in the MANCOVA comparing the samples (e.g., no interaction effect group*gender). Therefore, the methods and especially the results have changed considerable. 

* It's unclear why these extra items were generated "In addition, items concerning the subjective skill to handle anxiety and the acceptance of anxiety were generated and all but one were worded without negations" as well as the extra 30th item. Were these items included in the analyses?

If so, which items were they?

Answer: The extra items were generated following an exploratory approach as an addition to the model, focusing on two other ER approaches (acceptance and the subjective feeling to be able to control emotions). Due to the item analysis these items were removed even before the factor analysis because of low discriminatory power or because the lowered overall internal consistency. (lines 313ff)

* It's strange that the CFA was just mentioned in passing in the method section rather than given a full section in the results with all the appropriate information provided.

Answer: Thank your noticing this. In the new structure, the use of EFA and CFA is clearer (see overall comment for methods and results). 

* The differences in administration and characteristics between the two samples are a little concerning. Ideally the samples would be analyzed separately and assessed for any differences between them, or at least this would be done and noted in a footnote if no differences were found.

Answer: Thank you for pointing this out. We agree that the use of two samples combined into one was rather confusing. Following your suggestion, we have redone the analyses with separate samples (see overall comment for methods and results).

* Were all participants given potential reading assistance or only in the second sample where it is noted?

Answer: Yes, in both groups, assistance was offered. I have changed the wording, so that it is clearer for the reader. (line 206)

* The confirmatory factor analysis is mentioned again in passing in the statistical analysis section, and described as unsuccessful--this is confusing, and once again, more detail should be provided about these analyses (at the very least, more of the fit statistics, and an indication of exactly what model was tested). A little more detail is provided in the results section but it's unclear each time it's mentioned previously that it will be discussed later in the paper and even when it comes up in the results section more information should be provided. It's an unusual strategy to do the CFA first and then do an EFA when the CFA fails. Usually an EFA would be conducted first and the CFA to confirm the structure that was uncovered. Although the measure was developed based on theory, it still seems the traditional analytic process would make more sense. The CFA should also generally be done in a separate sample than the EFA. So, ultimately, it might make more sense to leave the CFA out of this paper entirely and conduct a follow up study to confirm the EFA structure in a new sample.

Answer: Thank you, once again, for pointing out the problem with the factor analysis. Please see the overall comment for the methods and results. 

Results

* More information should be provided on the EA in the determining of the number of factors - numeric results from the parallel analysis, the scree plot, the Eigenvalues and % of variance explained, etc.

Answer: Thank you for this recommendation. I have added numeric details of the parallel analysis, as well as Eigenvalues of the data and explained variance per factor to the manuscript. As far as I know this is usually not reported in detail in the manuscript, so I have added the information to the electronic supplement. If you would like to see the table in the actual manuscript, I would be glad to move it. (ESM table 2)

* Exactly how much would the overall reliability decline if item 1 was removed, because ideally items with low factor loadings should not be retained?

Answer: In the new analysis item 1 was removed because of a low factor loading. 

* It's unclear whether item 9 was retained on both subscales. 

Answer: In the new analysis item 9 only loaded satisfyingly on one subscale. 

* Item 23 does not seem to make sense on the reappraisal subscale and may need to be removed

Answer: It seems the children interpreted the item differently than we have thought. After discussing this in the author team we have decided to keep it on the reappraisal subscale. The children in the CAD group seem to have interpreted the item as that they ignore what makes them afraid (and think of something positive instead). We have emphasized the discrepancy in the discussion and see need for further research here. (lines 452ff)

* Although it states in text this information is in Table 2, I don't see the factor intercorrelations in the table.

Answer: Thank you for noticing this mistake. The information on factor intercorrelations can be found in the electronic supplements 

* The lack of correlations with the SDQ internalizing should be mentioned under the convergent validity header (as that was their intention), not divergent. Thus, convergent validity is only partially confirmed. Similarly, the correlations with adaptive subscale should be listed under divergent validity. There is a problem with the divergent validity evidence, as there is not evidence of divergent validity for this scale.

Answer: Thank you for bringing this to our attention. In the new analysis the internalizing and externalizing as well as adaptive and maladaptive scale correlations are labeled correctly as convergent or divergent validity.

* It's unclear why some analyses were done with the samples separated and some were not.

Answer: Thank you for this comment. In the new analysis this should be clearer (see overall comment for methods and results).

* It seems like it should say p < .001, not p > .001.

 Thank you for noticing this mistake. I have corrected the wrong symbols. 

Discussion

* Unfortunately, I would not agree that the construct has been confirmed, given the poor divergent evidence. Ultimately though, the biggest issue is there are not enough validity measures in the study, so it's difficult to draw conclusions. There should be validity measures that are more other existing emotion regulation and strategy questionnaires, not just validity measures of psychopathology. It's one thing to show relations to externalizing problems where none were expected - as the authors state, ER is a transdiagnostic factor that is related to a lot of pathology - however, avoidance should be less related to externalizing, so it's still concerning. However, the correlations with adaptive coping are more concerning, given the lack of other relations to other ER measures among which to contextualize this.

Answer: Thank you for bringing this to our attention. We have added these concerns in the discussion. The new results are similar regarding the adaptive ER strategies correlations. We have reframed the results and limited it, especially for construct validity. Nonetheless, the results for factor structure and reliability, as well as some parts of construct and criterion validity are promising. We have tried to bring this discrepancy to the reader’s attention and emphasized a need for more research to clarify this, especially with other ER questionnaires (e.g., lines 540ff)

* The authors should interpret why the security scale possibly ended up splitting into two separate factors.

Answer: Thank you for this helpful comment. We think that the factor split up in a more active and a more passive component of situation modification. We have added a sentence to the discussion. (lines 461ff)

* There should be a more thorough discussion of limitations.

Answer: Thank you for noticing this. In accordance with the new results, we have completely reworked the limitations. (lines 525ff)

Thank you once again for this fair and thorough review. It helped us a lot to improve our work.

---

## [Decision Letter · Decision Letter 1]

30 May 2022

PONE-D-21-23200R1Bochum Assessment of Avoidance-based Emotion Regulation for Children (BAER-C): Development and Evaluation of a New Instrument measuring Anticipatory Avoidance-based Emotion Regulation in anxiety eliciting situationsPLOS ONE

Dear Dr. Lippert,

Thank you for submitting your manuscript to PLOS ONE. After careful consideration, we feel that it has merit but does not fully meet PLOS ONE’s publication criteria as it currently stands. Therefore, we invite you to submit a revised version of the manuscript that addresses the points raised during the review process. The Reviewer and I see many improvements to the manuscript based on the initial set of comments. The Reviewer provides several additional suggestions for you to consider to make your analyses and interpretation of the results more rigorous. Please consider each of the suggestions carefully. I anticipate that I should not need to send your revision for further review, but will review the revision myself.

We look forward to receiving your revised manuscript.

Kind regards,

Thomas M. Olino

Academic Editor

PLOS ONE

Journal Requirements:

Reviewers' comments:

Reviewer's Responses to Questions

**Comments to the Author**

1. If the authors have adequately addressed your comments raised in a previous round of review and you feel that this manuscript is now acceptable for publication, you may indicate that here to bypass the “Comments to the Author” section, enter your conflict of interest statement in the “Confidential to Editor” section, and submit your "Accept" recommendation.

Reviewer #1: (No Response)

2. Is the manuscript technically sound, and do the data support the conclusions?

Reviewer #1: Yes

3. Has the statistical analysis been performed appropriately and rigorously? 

Reviewer #1: Yes

4. Have the authors made all data underlying the findings in their manuscript fully available?

Reviewer #1: Yes

5. Is the manuscript presented in an intelligible fashion and written in standard English?

Reviewer #1: Yes

6. Review Comments to the Author

Reviewer #1: The authors did a commendable job in revising the paper in line with the reviewer and editor feedback. I appreciate how responsive they were, and I feel the analyses are greatly improved and clarified. I am pleased with the new name the authors have given to the scale and to the paper, which greatly clarifies the target of this approach. I think the authors have also modified their conclusions and recommendations reasonably in the discussion. Below I have several further suggestions for clarifications in the revised paper, regarding the new analyses primarily. These suggestions, though numerous, are fairly straightforward and I am confident the authors can address them.

- Although the items were later dropped, it’s still not clear why these additional items (subjective skill, acceptance of anxiety) were generated since they don’t fit with the model. This information should be provided because the reader doesn’t know yet when they are mentioned that they will later be dropped, so it’s still confusing to read. Another idea could be to relegate this information to a footnote later on so it doesn’t take on such a high priority when describing the scale – but even in that case more information should be provided about why these items were developed.

- The statement about “for scoring…” that is on page 10 line 197 of the tracked changes copy should be moved to after the results section because we can’t determine how the questionnaire should be scored until the factor structure information is presented.

- For validity, I would refer to it as construct validity, not criterion validity (as it’s not being compared to a gold standard measure of the same construct). Also, it would be useful to provide validity information in both samples separately (and then together, if desired).

- Low reliability of SDQ scales should be mentioned in the limitations as a potential issue with attenuating correlations.

- Please provide examples of the maladaptive and adaptive strategies on the FEEL-KJ. Considering that in the results the measure was correlated positively with both adaptive and maladaptive scales, it would be helpful to contextualize why that might be by knowing more about what distinguishes the items.

- On line 378, I’m not sure why it says the differences between the samples were used to ‘check for criterion validity’.

- What is meant by three items were removed because of low discriminatory power? The way the results section reads, this happened before the factor analysis, so is this based on something other than factor loadings?

- More specific information about how much each removed item lowered the overall consistency (or at least what it was before and after the three items were removed) should be provided.

- It is my experience that eigenvalue and % of variance information is generally presented in the manuscript itself, so that’s ideally where I would like to see it – but I’ll defer to the editor if they feel this material is best suited for supplementary materials.

- In the eigenvalue table, the values for factor 6 should be provided so that it’s clear the simulated data eigenvalue is higher than the original at that point.

- The scree plot should be provided.

- The eigenvalue data is not necessarily all pointing to the same conclusion – which is not to say that I disagree with the five factor solution, especially in light of the parallel analysis and the % of variance explained – but it’s also noteworthy that there are three factors with eigenvalues above 1, and there’s a huge drop between factors 1 and 2 (though hard to tell in proportion without the scree plot), so a 1 or 3 factor solution would also be defensible. These potential solutions and the decision making process justifying going with the five factor solution should be described in text.

- The loadings for item 1 and 25 should be provided even though they were removed (and can be italicized or otherwise indicated in the table if necessary for clarity).

- There shouldn’t be model fit statistics TLI and RMSEA provided for an exploratory factor analysis.

- It isn’t indicated that the authors intend users to create an overall total score across all 21 items, so if this doesn’t make conceptual sense, then an overall alpha shouldn’t be provided across all 21 items (or if it is intended, this should be clarified).

- Is the validity with the SCAS calculated combining both samples? Since the other measures are only in one sample or the other, this measure should be analyzed separately for each sample.

- Do the SCAS-C and SDQ go in opposite directions, in terms of how they’re scored? Please clarify in the methods section how they’re coded because in Table 4 it’s unclear why the social reassurance scale is correlated in opposite directions with the SCAS-C and the SDQ internalizing…and also whether the correlations with suppression and reappraisal indicate higher levels of externalizing problems.

- In the divergent validity section, it should also say that social reassurance did not correlate significantly with externalizing symptoms.

- Please also report the chi square value for the CFA model.

- The loadings in the CFA should also be reported, either in text, a table, or a figure, whatever is preferable. A figure would be ideal so the structure can be visualized. Were the intercorrelations between the factors modeled (since they were accounted for in the EFA model)?

- The comparison between the two samples (3.7) should be presented earlier, before the rest of the results – probably when the rest of the sample characteristics are presented. Section 3.8 should stay where it is.

- I think there’s a typo in line 567 “trying that someone is with the child” – maybe there’s a word missing?

- In the discussion when the CAMS is referenced, it’s unclear how that’s relevant to a specific discussion of internal consistency.

- The statement in the discussion that all avoidance strategies except behavioral were associated with externalizing symptoms does not appear consistent with the results.

7. PLOS authors have the option to publish the peer review history of their article (what does this mean?). If published, this will include your full peer review and any attached files.

Reviewer #1: No

---

## [Author Response · Author response to Decision Letter 1]

11 Jul 2022

Dear Prof. Olino, dear Reviewer, 

Thank you once again for taking the time to review our manuscript. We deeply appreciate your recognition of the effort put into the revision so far. We have, once again, tried to answer and solve each of your comments. We hope all changes are to your satisfaction. 

Reviewer Comments

 Although the items were later dropped, it’s still not clear why these additional items (subjective skill, acceptance of anxiety) were generated since they don’t fit with the model. This information should be provided because the reader doesn’t know yet when they are mentioned that they will later be dropped, so it’s still confusing to read. Another idea could be to relegate this information to a footnote later on so it doesn’t take on such a high priority when describing the scale – but even in that case more information should be provided about why these items were developed.

Thank you for this remark. We have added information and literature to the method section that explains these items a bit more. (lines 180ff)

- The statement about “for scoring…” that is on page 10 line 197 of the tracked changes copy should be moved to after the results section because we can’t determine how the questionnaire should be scored until the factor structure information is presented.

Thank you for this helpful comment, of course, this is correct! We have moved the paragraph to the end of the factor analysis section. (lines 379ff)

- For validity, I would refer to it as construct validity, not criterion validity (as it’s not being compared to a gold standard measure of the same construct). Also, it would be useful to provide validity information in both samples separately (and then together, if desired).

Thank you for this comment. Criterion validity was changed to construct validity throughout the manuscript. The results regarding the validity are presented separately for each sample, only the SCAS was used in both samples. To avoid confusion regarding the use of the questionnaires in one sample or both samples, we have chosen not to present data of both samples combined. 

- Low reliability of SDQ scales should be mentioned in the limitations as a potential issue with attenuating correlations.

Thank you for this comment to further improve the manuscript. A sentence was added to the discussion section. (lines 542f)

- Please provide examples of the maladaptive and adaptive strategies on the FEEL-KJ. Considering that in the results the measure was correlated positively with both adaptive and maladaptive scales, it would be helpful to contextualize why that might be by knowing more about what distinguishes the items.

Thank you for noting this. We have added the subscales to the method section. (lines 317ff)

- On line 378, I’m not sure why it says the differences between the samples were used to ‘check for criterion validity’.

Thank you for this helpful clarification. It was changed to construct validity. 

- What is meant by three items were removed because of low discriminatory power? The way the results section reads, this happened before the factor analysis, so is this based on something other than factor loadings?

This is correct. Before exploratory factor analysis items were analyzed for item difficulty and discriminatory power. Those three items were removed because discriminatory power was <.30. 

- More specific information about how much each removed item lowered the overall consistency (or at least what it was before and after the three items were removed) should be provided.

Thank you for pointing this out. The alpha value increased from .88 to .91. In the analysis we acted conservatively and dropped any item, which would increase alpha. Due to the dropping of the overall alpha value following your other comment, we have included those items in the factor analysis. All three items dropped out because they failed to load on any of the five factors. We have changed this in the results section. (lines 373f)

- It is my experience that eigenvalue and % of variance information is generally presented in the manuscript itself, so that’s ideally where I would like to see it – but I’ll defer to the editor if they feel this material is best suited for supplementary materials.

- Thank you for once again clarifying this. Dear Editor, would it be possible to clarify where you think this data should be in the manuscript? 

- In the eigenvalue table, the values for factor 6 should be provided so that it’s clear the simulated data eigenvalue is higher than the original at that point.

- Thank you for this remark. Data for the 6th factor was added to the manuscript.(Table ESM 2) 

- The scree plot should be provided.

- The scree plot was added to the electronic supplements.

- The eigenvalue data is not necessarily all pointing to the same conclusion – which is not to say that I disagree with the five factor solution, especially in light of the parallel analysis and the % of variance explained – but it’s also noteworthy that there are three factors with eigenvalues above 1, and there’s a huge drop between factors 1 and 2 (though hard to tell in proportion without the scree plot), so a 1 or 3 factor solution would also be defensible. These potential solutions and the decision making process justifying going with the five factor solution should be described in text.

Thank you for this helpful comment. We have added additional explanations to the discussion section of the paper, discussing possible one and three factor solutions. (lines 506ff)

- The loadings for item 1 and 25 should be provided even though they were removed (and can be italicized or otherwise indicated in the table if necessary for clarity).

- The loadings were added to the table and following your suggestion were italicized. (Table 2)

- There shouldn’t be model fit statistics TLI and RMSEA provided for an exploratory factor analysis.

Thank you for this comment. The editor asked for TLI and RMSEA values for the exploratory analysis. So, I would keep it in the hands of the editor if those values should be removed. 

- It isn’t indicated that the authors intend users to create an overall total score across all 21 items, so if this doesn’t make conceptual sense, then an overall alpha shouldn’t be provided across all 21 items (or if it is intended, this should be clarified).

Thank you for this remark to clarify the manuscript. We have deleted the overall Alpha value but have kept the Omega as this uses the factor structure of the EFA. We think the results are much clearer now.

- Is the validity with the SCAS calculated combining both samples? Since the other measures are only in one sample or the other, this measure should be analyzed separately for each sample.

- Thank you for pointing this out. The SCAS validity was calculated using the anxiety sample. We have not reported the correlation for the student sample so far. We have added it to the table with the FEEL-KJ correlations (see table 5) 

- Do the SCAS-C and SDQ go in opposite directions, in terms of how they’re scored? Please clarify in the methods section how they’re coded because in Table 4 it’s unclear why the social reassurance scale is correlated in opposite directions with the SCAS-C and the SDQ internalizing…and also whether the correlations with suppression and reappraisal indicate higher levels of externalizing problems. 

- Both questionnaires are scored similarly (from 0 to 2 and from 0 to 3) so we think this does not explain the correlation. Nonetheless we have added the information to the method section (lines 278 & 297.). 

- In the divergent validity section, it should also say that social reassurance did not correlate significantly with externalizing symptoms.

- Thank you for this comment. We have changed the paragraph in the divergent validity section (line 419).

- Please also report the chi square value for the CFA model.

- The chi square value was added to the paper. The chi square test is significant. Following the recommendations of and Alavi and colleagues (2020) referring to Cole (1987) we have calculated the ratio ChiSqaure/df as it is more reliable regardless of the sample size. The ratio is 1.54 and thus below 2, therefore showing acceptable model fit especially in combination with the other values. We have added all information to the paper, as we have neglected them so far to keep the paper shorter (lines 427ff).

- The loadings in the CFA should also be reported, either in text, a table, or a figure, whatever is preferable. A figure would be ideal so the structure can be visualized. Were the intercorrelations between the factors modeled (since they were accounted for in the EFA model)?

Thank you for the remark. A figure visualizing the model was added to the results section. The intercorrelations were accounted for in the CFA model as well. 

- The comparison between the two samples (3.7) should be presented earlier, before the rest of the results – probably when the rest of the sample characteristics are presented. Section 3.8 should stay where it is.

- Thank you for this clarifying comment. We have moved the section to the methods section right after the sample characteristics. (lines 236ff)

- I think there’s a typo in line 567 “trying that someone is with the child” – maybe there’s a word missing?

- Thank you for notifying. We have corrected the sentence. (line 525)

- In the discussion when the CAMS is referenced, it’s unclear how that’s relevant to a specific discussion of internal consistency.

- Thank you for this comment. We have clarified the paragraph that was supposed to show that the internal consistency of the BAER-C is comparable to other questionnaires in the field. (lines 529f)

- The statement in the discussion that all avoidance strategies except behavioral were associated with externalizing symptoms does not appear consistent with the results.

Thank you for the remark. The part in the discussion was corrected to match the result section and once again suggesting problems especially with the suppression subscale. (lines 556ff)

---

## [Decision Letter · Decision Letter 2]

13 Oct 2022

PONE-D-21-23200R2Bochum Assessment of Avoidance-based Emotion Regulation for Children (BAER-C): Development and Evaluation of a New Instrument measuring Anticipatory Avoidance-based Emotion Regulation in anxiety eliciting situationsPLOS ONE

Dear Dr. Lippert,

Thank you for submitting your manuscript to PLOS ONE. After careful consideration, we feel that it has merit but does not fully meet PLOS ONE’s publication criteria as it currently stands. Therefore, we invite you to submit a revised version of the manuscript that addresses the points raised during the review process. Specifically,Language issuesPLOS ONE author guidelines should be followedPresentation tables, Figure, and References.Manuscript should be adhered to standard checklist.Please submit your revised manuscript by Nov 27 2022 11:59PM. If you will need more time than this to complete your revisions, please reply to this message or contact the journal office at plosone@plos.org. Please include the following items when submitting your revised manuscript:A rebuttal letter that responds to each point raised by the academic editor and reviewer(s). You should upload this letter as a separate file labeled 'Response to Reviewers'.A marked-up copy of your manuscript that highlights changes made to the original version. You should upload this as a separate file labeled 'Revised Manuscript with Track Changes'.An unmarked version of your revised paper without tracked changes. You should upload this as a separate file labeled 'Manuscript'.If applicable, we recommend that you deposit your laboratory protocols in protocols.io to enhance the reproducibility of your results. Protocols.io assigns your protocol its own identifier (DOI) so that it can be cited independently in the future. For instructions see: https://journals.plos.org/plosone/s/submission-guidelines#loc-laboratory-protocols. Additionally, PLOS ONE offers an option for publishing peer-reviewed Lab Protocol articles, which describe protocols hosted on protocols.io. Read more information on sharing protocols at https://plos.org/protocols?utm_medium=editorial-email&utm_source=authorletters&utm_campaign=protocols.

We look forward to receiving your revised manuscript.

Kind regards,

Nabi Nazari, PhD

Academic Editor

PLOS ONE

Journal Requirements:

Additional Editor Comments (if provided):

Dear Authors,

Please, revise the manuscript according following comments.

First: Please, follow the Author guideline.

■ in text citation, use Square bracket [ ], with numbers.

■ add Doi to references.

The quality of language is acceptable. However, there are several similar issues.

Unnecessary use of prepositions at the beginning or between sentences has reduced the writing quality of the article.

"Examples are given below. Please revise the entire text according to these examples. You can also use author services".

I suggest you that consult with professional editor.

Please, delete numerous unnecessary prepositions or words.

1. Line 51: please revise the sentence. " with Period" is unnecessary and not suitable.

2. Line 61: " Even SO" is unnecessary. Also 20 to 40% is not correct. "20% to 40% " is correct.

3. Line 63: "still" is unnecessary. " Lack" needs article( 'the lack is correct).

4.line 73: authors state" numerous ER strategies". Authors should nake example.

5. Line 77: " Gross (2001)" delete. the citation should moved to the end of the sentence.

6. Line 80: Depete "especially" .

7. Line 80- 85: the paragraph is not relevant, short, and irrelevant to the befor and next Paragraphs.

7. Line 80. The" :" is not suitable.

8. Line 86: delete " albeit".

.......................

Following issues are Minor. However, some items are critical.

1. Add legend for tables and figure.

2.The abstract is not informative. Add some statisticall results.

3. Please follow IMRAD. the limitations is correct. Outlook should be deleted.

4. Please, speak about generalizability.

5. The study should be written according to " standard checklist". ● this item is critical. Please attach the STROBE CHECKLUST.

6. The figure should be presented in standard form.

The current figure is not scientific.

Reviewers' comments:

Reviewer's Responses to Questions

**Comments to the Author**

1. If the authors have adequately addressed your comments raised in a previous round of review and you feel that this manuscript is now acceptable for publication, you may indicate that here to bypass the “Comments to the Author” section, enter your conflict of interest statement in the “Confidential to Editor” section, and submit your "Accept" recommendation.

Reviewer #1: (No Response)

Reviewer #2: (No Response)

2. Is the manuscript technically sound, and do the data support the conclusions?

Reviewer #1: Yes

Reviewer #2: Partly

3. Has the statistical analysis been performed appropriately and rigorously? 

Reviewer #1: Yes

Reviewer #2: Yes

4. Have the authors made all data underlying the findings in their manuscript fully available?

Reviewer #1: Yes

Reviewer #2: Yes

5. Is the manuscript presented in an intelligible fashion and written in standard English?

Reviewer #1: Yes

Reviewer #2: Yes

6. Review Comments to the Author

Reviewer #1: The authors have been very responsive to the reviews, and I think the scale will make an important contribution to the literature. I believe the paper should be accepted for publication, with the following last few minor suggestions for improvement:

• The information justifying the acceptance and subjective skill is good, but the end of the paragraph is still confusing. It still says four subscales with 29 items were created (and then a 30th item), so it’s not clear where the acceptance and subjective skill items went. It looks like the 4 subscales covered 25 items, based on the tables – so I guess I would put the statement ‘four subscales with 25 items were created’ before the information about acceptance and subjective skills, and then state that an additional four items were added to cover these topics.

• On line 185 of the tracked changes copy, the word importance doesn’t seem correct. I’m not sure what word is supposed to be there, but that word doesn’t make sense.

• You note that validity is examined separately, but on line 216 it states “Finally, to check construct validity both patient and community sample were combined.” I think you mean to say validity was examined in both the patient and community samples?

• If the validity correlations presented really are for different samples depending on the measure, it should be made more clear which sample each result is coming from (stated in the results when each set of results is presented).

• When I said it wasn’t clear how testing differences between the samples would speak to validity, I meant it should be made more clear what the expectation is and how it would speak to validity (e.g., if scores were higher in the clinical sample this would indicate that the scale accurately measured avoidance based emotion regulation). And actually, that one could be considered criterion validity! I think you can change that particular instance back (or leave it, whatever you prefer).

• The explanation added to the discussion regarding choosing the five factor solution over the one or three factor solution should be in the second paragraph of the results section instead.

• The CFA figure is very helpful, thank you! One suggestion would be to make the oval for the reappraisal figure bigger, and spell out that word, so that it’s clear the reappraisal and avoidance factors are on the same hierarchical level and correlated with each other (I do see the double headed arrow but the size and orientation make it easy to miss) rather than reappraisal appearing as another lower level factor under avoidance.

Reviewer #2: Dear Authors,

Thank you for designing this precious study.

1. That is suggested to provide a short description for “anxiety disorders” in first paragraph in INTRODUCTION section.

2. It should be better change “ criterion validity” to “ construct validity”.

3. Please mention examples of the maladaptive and adaptive strategies on the FEEL-KJ.

4. Please more explain why removed three items.

5. chi square value for the CFA model is not reported!

Sincerely

7. PLOS authors have the option to publish the peer review history of their article (what does this mean?). If published, this will include your full peer review and any attached files.

Reviewer #1: No

Reviewer #2: No

---

## [Author Response · Author response to Decision Letter 2]

9 Dec 2022

Dear Dr. Nazari, dear Reviewers, 

Thank you once again for taking the time to review our manuscript. We deeply appreciate your recognition of the effort put into the revision so far. We have, once again, tried to answer and solve each of your comments and to respect all author guidelines. We hope all changes are to your satisfaction. 

Editor Comments 

First: Please, follow the Author guideline.

■ in text citation, use Square bracket [ ], with numbers.

■ add Doi to references.

All references have been updated and changed. I apologize for overlooking. 

Unnecessary use of prepositions at the beginning or between sentences has reduced the writing quality of the article. I suggest you that consult with professional editor.

Thank you for your thorough work and this helpful comment. As recommended a copy editor from the field of clinical psychology re-checked the manuscript. We have removed or rewritten all sentences in which the prepositions could be removed without changing the meaning of the sentence. We think the manuscript is a lot tighter now. 

Following issues are Minor. However, some items are critical.

1. Add legend for tables and figure.

All legends for tables and figures have been adapted to author guidelines and example manuscript. Table 3 was intentionally left without legend. 

2.The abstract is not informative. Add some statisticall results.

Thank you for this helpful comment, I have added some statistical results to the abstract. 

3. Please follow IMRAD. the limitations is correct. Outlook should be deleted.

Thank you for noting that detail. Outlook is deleted. 

4. Please, speak about generalizability.

Thoughts about generalizability were added to the discussion. (lines 570-577)

5. The study should be written according to " standard checklist". ● this item is critical. Please attach the STROBE CHECKLUST.

The STROBE Checklist was attached and filled out according to the STROBE article. Thank you for bringing this to our attention. The Checklist really helped to tighten the structure of the paper. Very few points were not applicable because of the structure of this study introducing a new questionnaire. 

6. The figure should be presented in standard form.

The current figure is not scientific.

 We have changed the figure according to the journal guidelines. 

Reviewer 1 Comments

The authors have been very responsive to the reviews, and I think the scale will make an important contribution to the literature. I believe the paper should be accepted for publication, with the following last few minor suggestions for improvement:

The information justifying the acceptance and subjective skill is good, but the end of the paragraph is still confusing. It still says four subscales with 29 items were created (and then a 30th item), so it’s not clear where the acceptance and subjective skill items went. It looks like the 4 subscales covered 25 items, based on the tables – so I guess I would put the statement ‘four subscales with 25 items were created’ before the information about acceptance and subjective skills, and then state that an additional four items were added to cover these topics.

Thank you for this helpful comment to further clarify the manuscript. I have followed your suggestions and changed the paragraph according to it. (lines 173 ff)

On line 185 of the tracked changes copy, the word importance doesn’t seem correct. I’m not sure what word is supposed to be there, but that word doesn’t make sense.

Thank you for pointing this out, The mistake was corrected. 

You note that validity is examined separately, but on line 216 it states “Finally, to check construct validity both patient and community sample were combined.” I think you mean to say validity was examined in both the patient and community samples?

Thank you once again. I have removed the misleading sentence, as the process was explained better in the paragraph already.

If the validity correlations presented really are for different samples depending on the measure, it should be made more clear which sample each result is coming from (stated in the results when each set of results is presented).

Thank you for this remark. I have clarified this in the headings of the paragraphs. (lines 394, 411)

When I said it wasn’t clear how testing differences between the samples would speak to validity, I meant it should be made more clear what the expectation is and how it would speak to validity (e.g., if scores were higher in the clinical sample this would indicate that the scale accurately measured avoidance based emotion regulation). And actually, that one could be considered criterion validity! I think you can change that particular instance back (or leave it, whatever you prefer).

Thank you for clarifying. I think I leave it the way it is now as it is clear in its structure. 

The explanation added to the discussion regarding choosing the five factor solution over the one or three factor solution should be in the second paragraph of the results section instead.

Following your recommendations, the paragraph was moved to the results section. Thank you.

The CFA figure is very helpful, thank you! One suggestion would be to make the oval for the reappraisal figure bigger, and spell out that word, so that it’s clear the reappraisal and avoidance factors are on the same hierarchical level and correlated with each other (I do see the double headed arrow but the size and orientation make it easy to miss) rather than reappraisal appearing as another lower level factor under avoidance.

Thank you for this comment. The figure was changed according to the editors comments. I have tried to include your feedback as well. 

Reviewer 2 Comments

Thank you for designing this precious study.

1. That is suggested to provide a short description for “anxiety disorders” in first paragraph in INTRODUCTION section.

Thank you for your suggestion. I have added a very short DSM-5 TR definition to the introduction section. (lines 54f)

2. It should be better change “criterion validity” to “ construct validity”.

I have checked the whole manuscript and was not able to find that “criterion validity” was still in use. I would kindly ask you, to tell me the lines, in which you have found “criterion validity”.

3. Please mention examples of the maladaptive and adaptive strategies on the FEEL-KJ.

Paragraph 2.5.4 describes the questionnaire and lists all strategies which load on the adaptive and the maladaptive second order factor. Please tell me, if you would like example for the strategies in other paragraphs as well. To keep the paper short, I would prefer to leave it in this paragraph. 

4. Please more explain why removed three items.

Thank you for this remark. The items were removed because their discriminatory power was too low (below the threshold of .30). This is stated in the first section of the results. 

5. chi square value for the CFA model is not reported!

Thank you for this remark. The χ²-value was added to the results section.

---

## [Editor Report · Decision Letter 3]

13 Dec 2022

Bochum Assessment of Avoidance-based Emotion Regulation for Children (BAER-C): Development and Evaluation of a New Instrument Measuring Anticipatory Avoidance-based Emotion Regulation in Anxiety Eliciting Situations

PONE-D-21-23200R3

Dear Dr. Lippert,

We’re pleased to inform you that your manuscript has been judged scientifically suitable for publication and will be formally accepted for publication once it meets all outstanding technical requirements.

Kind regards,

Nabi Nazari, PhD

Academic Editor

PLOS ONE
---

## [Editor Report · Acceptance letter]

5 Jan 2023

PONE-D-21-23200R3 

Bochum Assessment of Avoidance-based Emotion Regulation for Children (BAER-C): Development and Evaluation of a New Instrument Measuring Anticipatory Avoidance-based Emotion Regulation in Anxiety Eliciting Situations 

Dear Dr. Lippert:

I'm pleased to inform you that your manuscript has been deemed suitable for publication in PLOS ONE. Congratulations! Your manuscript is now with our production department. 

Kind regards, 

on behalf of

Dr. Nabi Nazari 

Academic Editor

PLOS ONE